# Anomalous friction of supercooled glycerol on mica

Mathieu Lizée [1] ✉, Baptiste Coquinot [1], Guilhem Mariette [1], Alessandro Siria [1] & Lydéric Bocquet [1] ✉

Although friction of liquids on solid surfaces is traditionally linked to wettability, recent works have unveiled the role of the solid's internal excitations on interfacial dissipation. In order to directly evidence such couplings, we take advantage of the considerable variation of the molecular timescales of supercooled glycerol under mild change of temperature to explore how friction depends on the liquid's molecular dynamics. Using a dedicated tuning-fork AFM, we measure the slippage of glycerol on mica. We report a 100 fold increase of slip length upon cooling, while liquid-solid friction exhibits a linear scaling with molecular relaxation rate at high temperature. This scaling can be explained by a contribution of mica's phonons which resonate with density fluctuations in the liquid, allowing efficient momentum transfer to mica. These results suggest that engineering phononic spectra of materials could enhance flow performance in nanofluidic channels and industrially relevant membranes.

Solid-liquid friction is usually accounted for by the hydrodynamic slip length, defined as the ratio between the bulk viscosity of the fluid and the solid-liquid friction coefficient. The topic has been the subject of many investigations[1,2] based on a considerable body of experimental and theoretical work and it is now established that interfacial friction is determined to a large extent by surface energy, with slippage being promoted by hydrophobicity[1]. However, this picture is incomplete and fails to explain some experimental measurements on liquid–solid friction. In particular, it is insufficient to account for the radius-dependent and ultra-low friction observed in carbon nanotubes and graphene-based nanochannels[3–6], as well as the difference of friction inside nanotubes versus flat graphite[7]. Recently, and beyond the picture of an inert wall, the role of solid dynamics on interfacial friction was put forward. Numerical simulations revealed indeed that mechanical fluctuations of the confining wall do affect wetting, slippage and diffusive transport inside the liquid[8–10], while, conversely, several experiments have shown a strong influence of adsorbed liquids on the band structure and lifetime of phonons in the underlying solid[11,12]. But even more counterintuitively, the couplings of electronic −plasmon-like−excitations in metals with the collective charge fluctuations in liquids were shown to generate strong fluctuation-induced contributions to friction[13,14]. This electronic friction explains the peculiar radius dependency of liquid slippage in carbon nanotubes[4,13] and the anomalous electron cooling of graphene in contact with water[15]. Reversly, it was reported that liquid friction can induce electronic currents in graphene. This measurement demonstrates a momentum transfer from liquid flows to electrons which is believed to be mediated by acoustic phonons[16–18]. Interestingly, the emergence of couplings between the liquid fluctuations and the solid (electronic or phononic) excitations is reminiscent of the longstanding quest for electronic and phononic contributions to solid-on-solid friction. Recently, these dissipation channels were elegantly disentangled by showing a strong drop of non-contact friction at the superconducting transition of Nb thin films[19]. Independently, solid friction on graphene/mica substrates was shown to be enhanced by an intercalated water layer[20], via a broadening of the phonon spectrum of graphene by water. How these concepts extend to liquid–solid friction remains largely unknown, and despite first hints, the role of solid-state excitations−either phononic or electronic−on liquid-solid friction has never been measured directly and remains elusive. Such a tunable picture of the interface opens countless possibilities for nanoscale flow engineering via the controlled electronic properties of the channel's wall[18].

[1]Laboratoire de Physique de l'Ecole Normale Supérieure, ENS, Université PSL, CNRS, Sorbonne Université, Université de Paris, 75005 Paris, France.
✉e-mail: mathieu.lizee@ens.fr; lyderic.bocquet@ens.fr

In this study, we address this question from a reversed perspective: instead of studying liquid friction on a variety of solids with different phononic or electronic properties, we rather vary the excitation spectrum of the liquid. To this end, we leverage the dynamical slowdown of supercooled liquids whose molecular timescale typically follows the Vogel-Fulcher-Tammann law: $\tau \propto \exp(E_a/k_B(T - T_g))$ above their glass transition temperature $T_g$ (here $E_a$ is an activation energy—see SI Section I for details). For convenience, we use pure glycerol, which is supercooled close to room temperature. Cooling by a few tens of Kelvins causes a drastic redshift of glycerol's spectral modes, as measured previously in mechanical and dielectric spectroscopy[21–23]. Although a few experimental groups have reported temperature-dependent slippage measurements in aqueous electrolytes[24], polymers[25], or oils[26], we are not aware of any measurements for supercooled molecular liquids on flat surfaces. Exploring how the frequency shift in supercooled liquids molecular dynamics affects liquid–solid friction is the main objective of this experimental study.

Accordingly, we investigate the surface dynamics of supercooled glycerol on mica surfaces, in a range of temperature 0–35 °C, corresponding to a 10% change in absolute temperature. The cleaved (001) surface of muscovite mica has a honeycomb structure, with a molecularly smooth corrugation[27]. This atomic smoothness makes mica a sample of choice for surface force investigations of liquid interfaces, evidencing, e.g. a signature molecular structuration of the liquid. Interestingly, a no-slip condition (corresponding to an immeasurably high friction) was reported for various fluids on mica, in spite of the atomic smoothness of the interface[7,28]. Here we specifically designed a tuning-fork-based atomic force microscope (AFM) system allowing us to measure the slip length of glycerol, with nanometric resolution, as a function of temperature.

We will then put our experimental results into perspective with predictions of molecular dynamics (MD) simulations of model supercooled liquids. These simulations suggest an increased slippage in the supercooled regime of water, ethanol[29], or binary Lennard–Jones liquids[30], by a factor up to 5 when decreasing temperature. The authors further reported a decrease of friction as temperature increases, in agreement with an activated rate process for which enhanced thermal agitation accelerates liquid dynamics, thus reducing friction. Interestingly, however, these simulations use frozen walls and neither consider the internal degrees of freedom of the solids, nor their

couplings to the liquid. These numerical results thus act as benchmark predictions for surface friction without the solid internal dynamics.

## Results

Our measurements of liquid–solid friction are based on an axi-symetric flow between an AFM tip and a mica surface, as sketched in Fig. 1a–c. A tungsten tip of micrometric diameter is glued on the quartz tuning fork and immersed in a glycerol droplet placed on a freshly cleaved mica surface. The prong is excited at its resonant frequency of roughly 30 kHz with a piezo dither, leading to a vertical oscillatory motion of the tip. The tuning fork's oscillation signal is measured with a transimpedance preamplifier (FEMTO DLPCA-200). This signal is then directed to a phase-locked-loop (Nanonis Mimea) which keeps the phase to zero and the system at resonance (cf. inset of Fig. 1a) and monitor the excitation voltage to maintain a constant nanometric amplitude ($a = 3$ nm).

The drainage flow between the tip and mica induces forces that cause a slight change of the resonance curve: elastic forces that are in phase with displacement, shift the fork's resonance frequency by $\delta f$. On the other hand, viscous friction damps the oscillation. As the tip approaches the surface, the quality factor is reduced with respect to its long-distance value ($Q \sim 100$). This additional damping is compensated by an increase of the piezo excitation ($E$) to maintain the constant oscillation amplitude $a$. With this measurement scheme, we obtain the complex mechanical impedance $Z$ of the confined liquid at 30 kHz[31] (cf. Fig. 1c):

$$Z' + iZ'' = 2K_{\text{eff}}\frac{\delta f}{f_0} + i\frac{K_{\text{eff}}}{Q_0}\left(\frac{E}{E_0} - 1\right) \tag{1}$$

where $K_{\text{eff}}$ is the effective oscillator's stiffness, $f_0$, $Q_0$ and $E_0$ are, respectively, the free frequency, free quality factor and free excitation voltage. We stress that the very high stiffness ($\approx 40$ kN) of this system makes it very robust with respect to strong forces that typically cause the tip to snap-in to the surface in AFM experiments. Moreover, ensuring that only the very end of the tip is in glycerol yields an excellent quality factor and thus an exquisite resolution on $Z$.

The mechanical contact between the AFM probe and the mica surface is detected as a divergence of the frequency shift $\delta f$ (contact stiffness) shown in Fig. 1d. Further examples are shown in the SI-figure S3, illustrating that the discontinuity in the frequency-shift signal is

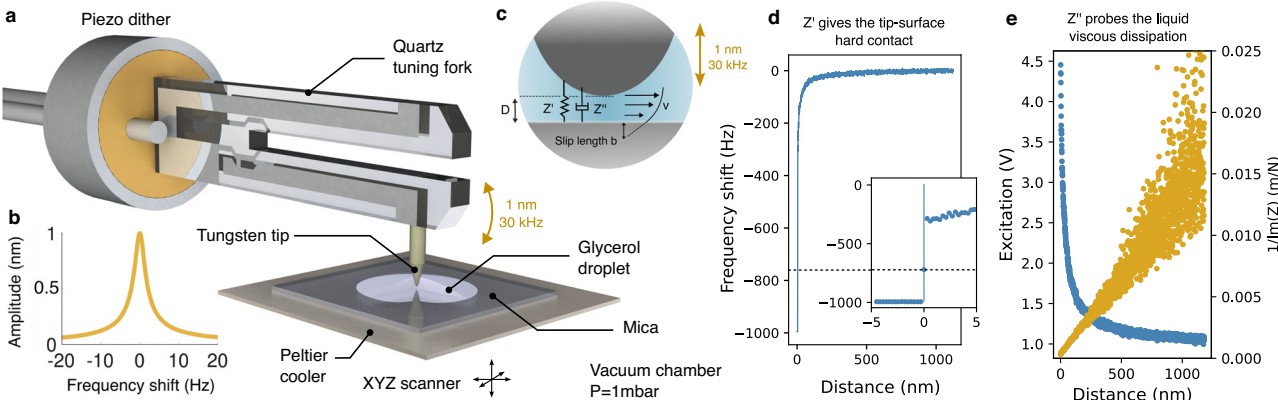

**Fig. 1 | AFM slippage measurement. a** Schematics of the tuning fork AFM system, with a typical resonance curve on (**b**). In (**c**), we sketch the drainage flow between the micrometric tungsten tip and the mica surface. If we extrapolate linearly the velocity profile in the solid. It vanishes at a non-zero distance from the wall: the slip length. The no-slip boundary condition ($b = 0$) thus corresponds to a zero velocity at the surface. On the other hand, a perfect slip boundary condition (zero friction) is equivalent to a vanishing of the stress at the interface. The AFM system allows to extract the complex mechanical impedance of the contact at 30 kHz $Z = Z' + Z''$ for a given sphere-plane distance $D$. **d** Elastic response $Z'$ of the confined liquid. The inset shows a zoom on the mechanical contact where the phase regulation is suddenly lost and the frequency shift saturates at −1 kHz. This discontinuity allows us to determine the position of the mechanical contact ($D = 0$) with nanometric resolution. **e** Typical approach curve at room temperature (blue) showing the dissipation increasing as more energy is lost to the confined viscous flow. In orange, we show the inverse dissipative impedance $1/Z''$ versus sphere-plane distance $D$.

measured in a systematic and perfectly reproducible manner. This divergence reliably provides the origin of the x-axis ($D = 0$) with nanometric resolution. In Fig. 1e, we further plot the excitation $E$ and inverse dissipative impedance $1/Z'$ as functions of the sphere-plane distance $D$ in an approach curve measured at room temperature.

The mica sample is placed on a Peltier element allowing to control its temperature. To avoid any condensation of water in the glycerol at the lowest temperatures, we place the whole system in a vacuum chamber under a pressure of 1 mbar, above the vapor pressure of glycerol but below that of water. The chamber can also be filled with dry nitrogen and the pumps turned off during measurements to decrease the mechanical noise level. All curves presented here have been measured with the same tip of 6.5 $\mu$m radius (see SEM image in Fig. S5a), immersed in the droplet for several weeks. Below its melting temperature $T_M = 18$ °C, glycerol is in the supercooled regime and its viscosity increases continuously over a dozen orders of magnitude when approaching the glass transition at $T_g \sim -87$ °C[22,32]. We used a cone-plane rheometer to assess the purity of our glycerol and found its viscosity is in excellent agreement with the literature[33], showing that our water contamination level is inferior to 0.5% in mass (cf. Section I.A in SI). Measurements of the complex shear modulus at 30 kHz show that glycerol's visco-elasticity is negligible for $T > 0$ °C (cf. Section I.A in SI). In other words, hydrodynamic and molecular timescales are well separated.

In the presence of a slip length b at the surface and in the sphere-plane geometry, the dissipative mechanical impedance, $Z'$ is reduced with respect to the no-slip case and takes the Reynolds form at long distance[34]

$$Z''_{Reynolds} = \Im(Z) = \frac{6\pi R^2 \eta \, \omega}{D + b}, \qquad (2)$$

where $1/Z'$ is a straight line. Note that we assume a no-slip boundary condition for the relatively rough tungsten tip. At short distances, $D \leq b$, $1/Z'$ deviates from linearity as slippage induces strong deviations from the Reynolds equation and a more general formula is needed.

$$Z'' = \frac{6\pi R^2 \eta \omega}{D} f^* \left( \frac{b}{D} \right) \qquad (3)$$

The expression for $f^*$ was previously derived in the literature[34,35] (cf. Section II.A in SI) and is used to fit our approach curves in a systematic way. We neglect the elastic deformations of the solid surface leading to elasto-hydrodynamic corrections, as mica does not deform under the flow in present conditions (cf. Section II.C in SI).

Let us first examine the high-temperature regime (here, close to room temperature). As shown in Fig. 2e, the experimental results are in very good agreement with the usual Reynolds prediction in Eq. (2): the $1/Z'$ versus $D$ line crosses the x-axis close to $D = 0$, showing that the slip length $b$ is vanishingly small.

Now, as the temperature decreases from 30 °C down to 1 °C, one observes that the general shape of the curves for $1/Z'$ versus $D$ slightly change and the standard Reynolds expression in Eq. (2) is not sufficient to describe the dissipation for lower temperatures. However, the generalized prediction in Eq. (3) is shown to be in good agreement with the experimental data, provided a non-vanishing slip length $b$ is introduced; see Fig. 2a–e. By fitting the dissipation curves (blue points in Fig. 2) with Eq. (3) (see details in SI Section II.A), one can accordingly extract the slip length $b$ over a wide range of temperatures. The model obtained from the fit is then plotted as a black solid line on both the $Z'$ (blue) and $1/Z'$ (orange) points. Clearly, Eq. (3) reproduces very precisely the distance-dependence of the inverse approach curves $1/Z'$. The error on the slip length from a single fit is of the order of 3 nm for all temperatures. As a side note, we verified that the measured value of the slip length does not depend on the oscillation amplitude of the sphere, see SI-fig. S4.

## Discussion

Gathering results, we plot in Fig. 3a the extracted slip length $b$ as a function of the glycerol temperature between 0 °C and 35 °C. We immediately notice a two orders of magnitude increase of slippage upon cooling with $b$ going from 0-2 nm at 35 °C to ~200 nm at 1 °C. We emphasize that such behavior is not expected and contrasts with theoretical expectations and MD simulations[1,30]. Indeed, the slip length $b = \eta/\lambda$ of simple liquids is expected to be independent of viscosity[1], in accordance with the few experiments we are aware of[34]. This is due to the similar linear dependence on the liquid's molecular timescale for surface friction $\lambda$ on the one hand and viscosity $\eta$ on the other. Accordingly, $b$ would be expected here to depend only weakly on temperature. Such a behavior is indeed observed in MD simulations far above the glass transition[29,30], where an increase at most by a factor 5 is measured. In our experiments we do in fact observe a saturation of the slip length as a function of temperature below 10 °C. However, the huge decrease in slip length measured for $T > 10$ °C defies expectations.

Digging further, we plot the experimental friction coefficient $\lambda(T)$ versus temperature on Fig. 3b—see SI Section I for viscosity measurements. The friction coefficient is shown to exhibit a non-monotonous dependence on temperature, with a minimum around 10 °C and an increase of $\lambda(T)$ for $T > 10$ °C.

It is interesting to compare these observations with existing measurements of temperature-dependent slippage, which were e.g. measured with polymer melts[25] as well as in simulations[29,30,36]. In both cases, the authors reported a very strong increase of slip length upon cooling, primarily induced by the rise of viscosity. For the friction coefficient $\lambda = \eta/b$, however, both experiments and simulations demonstrate a strong drop upon heating. When plotting $\lambda$ versus $1/T$, an exponential scaling was reported in experiments: $\lambda \sim \exp(E_a/k_B T)$ where $E_a$ is an activation energy of order 0.1 eV (10 kJ/mol)[25]. This scaling is indeed that of an activated rate process: as temperature is increased, tangential diffusion of liquid molecules in the surface's corrugated potential is facilitated, leading to a friction drop. Turning back to our measurements in Fig. 3b, it is now clear that the non-monotonous dependency of $\lambda$ on $T$ and especially the increasing tendency above 10 °C is in drastic contrast with such a thermally activated rate process.

One may also notice that the strong increase of friction with temperature above 10 °C, may seem disproportionate to the relatively small variation in thermal energy (~10%). However, even in this moderate temperature range, glycerol dynamics are varied by a factor of 30, in agreement with the drastic increase of viscosity from 0.5 to 12 Pa.s. This is best evidenced in the strong redshift of the characteristic $\alpha$-peak of glycerol $f_\alpha(T)$ upon cooling; see Supplementary Section I. Accordingly, in order to highlight the relationship between friction and glycerol dynamics, we convert temperature into the corresponding frequency $f_\alpha(T)$ on the top axis of Fig. 3b. This plot highlights two regimes: for $f_\alpha < 34$ MHz, friction is measured to be decreasing with the molecular frequency $f_\alpha$, down to $\lambda_0 \sim 20$ MPa.s/m. For $f_\alpha > 34$ MHz, however, the friction coefficient increases with the molecular frequency $f_\alpha$.

Such a behavior has neither been reported experimentally up to now, nor explained theoretically. It strongly differs from the predictions of MD simulations, with frozen confining walls[29,30]. One can explore various possible mechanisms to explain it.

### Strain rate dependency and non-linearities

For liquid slippage, non-linearities may take the form of a strain-rate dependency where the strain rate $\dot{\gamma} \sim \partial_z v_x$ is a typical frequency scale of the flow. Such non-linearities are believed to emerge when microscopic timescales and strain rate are of the same order of magnitude. In our experiments, the strain rate is not constant during the approach but we can estimate it roughly as $\dot{\gamma} \sim 2\pi f a/D \sim 10$ kHz. We know from molecular dynamic simulations that both viscosity and friction

**Fig. 2 | Temperature dependent friction. a–f** Inverse dissipative impedance $1/Z''$ as a function of sphere-plane distance $D$. The solid lines are fits using Eq. (3) for $T \in \{2.45, 5.8, 9.5, 14.3, 18.6, 30.3\}°C$ (see SI Section II.A for details on the fitting process).

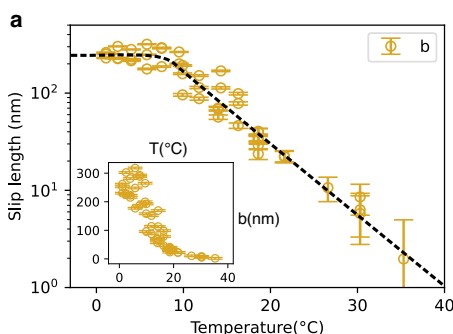
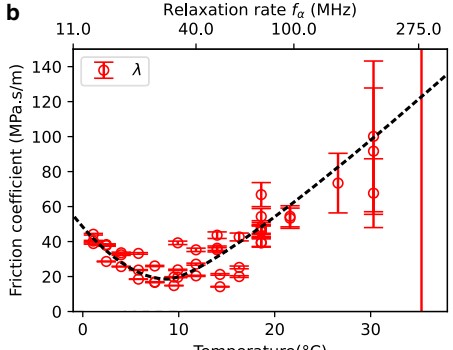

**Fig. 3 | Interfacial friction and slip length versus temperature.** All measurements are done in a single experimental run for various temperatures, with the same tip and on the same mica surface. **a** Slip length $b$ versus temperature with linear (inset) and log scales (main panel). Error bars correspond to 3 nm. **b** Resulting interfacial

friction coefficient, defined as $\lambda(T) = \eta(T)/b(T)$, versus temperature $\lambda(T)$. We also give the liquid's $\alpha$-relaxation frequency, $f_\alpha(T)$ on the top $x$-axis. Error bars correspond to a 3 nm error on the slip length. Source data are provided as a Source Data file.

coefficient are constant for $\dot{\gamma}\tau<0.1$ where $\tau$ is a typical liquid's relaxation rate[37]. In the worst case of $T = 0$ °C, this criterion yields a critical shear rate of roughly 100 kHz, more than one order of magnitude higher than what is reached experimentally. These considerations lead us to ignore the nonlinear effects caused by strain rate dependency of the friction coefficient.

**Interfacial water**

One may also consider the possibility that water or gas could condense preferentially at the hydrophilic mica surface. Considering the large affinity of glycerol with water, we were extremely careful to avoid water contamination: the 99.9% pure glycerol was deposited on freshly cleaved mica and immediately pumped down to 1 mbar in the

experimental cell for several days. Still, one can never completely rule out the possibility that water could condense at the mica surface. In such a case, this low viscosity lubrication layer at the mica interface would yield an apparent slippage proportional to the viscosity ratio, as $b_{app} \sim (\eta_{glycerol}/\eta_{water}) \times \delta_{water}$, with $\delta_{water}$ the thickness of the interfacial water layer[38]. The experimental results shown in Fig. 3 would imply a (temperature-dependent) thickness of the water layer in the range of $\delta_{water} \sim 10^{-2}$ nm, which is hardly realistic. We thus conclude that water segregation at the interface can be discarded as an underlying mechanism.

### Solid internal dynamics and fluctuation-induced dissipation

Putting our experimental results in perspective with the MD simulations of supercooled liquid friction—which report an Arrhenian behavior[29,30] and hence do not account for the non-monotonous dependence of friction with temperature, one key ingredient which is missing in the MD modeling is the internal dynamics of the solid. In contrast, recent MD calculation of liquid–solid friction including internal degrees of freedom of the solid showed that vibrational coupling may affect the friction coefficient[10]. It is therefore tempting to investigate the role of wall internal fluctuations on surface friction. Fundamentally, liquid friction can be calculated from the fluctuation-dissipation theorem using the Green-Kubo formula :

$$\lambda = \frac{1}{\mathscr{A} k_B T} \int_0^\infty dt \langle F_x(t) F_x(0) \rangle \tag{4}$$

with $\mathscr{A}$ the lateral area and $F_x$ is the fluid-solid interaction force. Let us assume that the fluid-solid interaction is described by an interaction potential $V_{FS}$. The surface force can be accordingly rewritten in terms of the (fluctuating) density distribution in the liquid ($n_l$) and the solid ($n_s$) as $F(t) = \int dr_s dr_l \nabla V_{FS}(r_s - r_l) n_s(r_e, t) n_l(r_w, t)$. Considering that fluctuations in the solid and in the liquid are uncorrelated at first order, we can rewrite the Green–Kubo equation in terms of the interfacial (2D) structure factors $S(r,r',t,t') = \langle n(r,t)n(r',t') \rangle$ of the two media. Going to Fourier space allows us to rewrite the friction coefficient in Eq. (4) as a function of the angle-averaged dynamical structure factors of the solid $S_s(q, \omega)$ and liquid $S_l(q, \omega)$:

$$\lambda = \frac{1}{8\pi^2 k_B T} \int_0^\infty dq\, V_{FS}(q)^2 q^3 \int_0^\infty d\omega\, S_s(q,\omega) S_l(q,\omega) \tag{5}$$

We now estimate $\lambda$ by identifying the key contributions to the solid and liquid fluctuation spectra. The full derivation is reported in the SI Section III and we discuss here the main steps.

**Liquid's structure factor.** Spectroscopy measurements of bulk glycerol—in particular dissipative mechanical and electrical impedance ($G''(\omega)$ and $\epsilon''(\omega)$)[22,39,40]—show that glycerol exhibits a non-dispersive Debye-like $\alpha$-peak at a temperature-dependent frequency $f_\alpha \sim e^{-E_a/k_B(T-T_g)}$, with $T_g$ the glass temperature transition. For the sake of simplicity and considering the dispersionless nature of the peak, we define an effective liquid's structure factor whose frequency dependence is Debye-like:

$$S_l(\omega) = \frac{\omega_\alpha(T)}{\omega_\alpha(T)^2 + \omega^2} \tag{6}$$

We refer to the Supplementary Information for further details ($q$ dependence, etc.).

**Solid's structure factor.** The solid's structure factor can be separated into a non-fluctuating (static) part that accounts for static corrugation and a dynamical part that accounts for internal fluctuations and excitations, $S_s = S_s^{static} + S_s^{dyn}$. This yields two complementary contributions

to friction: $\lambda = \lambda^{static} + \lambda^{dyn}$. The solid's static structure factor ($\omega = 0$) that accounts for roughness is strongly peaked at the reciprocal lattice typical wavevector $q_s^{max} \approx \frac{2\pi}{\sigma_s}$ with $\sigma_s$ a typical mica molecular length scale, so that

$$S_s^{stat}(q, \omega) \approx u_s \rho_s q_s^{max} \delta(q - q_s^{max}) \delta(\omega) \tag{7}$$

where $\rho_s \approx 1/\sigma_s^2$ is the atomic density on the interacting layer, $u_s$ is the amplitude of the roughness and $\delta$ the Dirac distribution. This simple expression for $S_s^{stat}$ reduces the molecular structure to the first peak in the liquid structure close to the surface, a feature which is the main contribution to the roughness-induced friction, see ref. 41. In the frequency domain, this static contribution is peaked at zero frequency. On the other hand, the dynamical part of the fluctuations spectrum, $S_s^{dyn}$, accounts for the internal excitations. As an insulator, mica does not exhibit low-energy electronic excitations, but phonon modes of mica were measured by Brillouin scattering[42]. Keeping the description as simple as possible, we model the dynamical structure factor of mica using one single acoustic phonon branch as

$$S_s^{dyn}(q, \omega) \equiv S_{ph}(q, \omega) = \pi \frac{T \rho_s}{mc^2} \delta(\omega \pm q \cdot c) \tag{8}$$

which is delta-peaked at the dispersion relation. Considering the aforementioned Brillouin scattering experiments, we estimate the sound velocity to be of the order of $c \approx 10^3$ m/s[42] while mica's lattice parameter is in the nanometer range[43].

**Frequency scalings and discussion.** Using these expressions for the spectra in Eq. (5), one can calculate the 'static' (corrugation induced) and 'dynamic' (solid fluctuation induced) contributions to the friction coefficient. Note that for the sake of simplicity and in order to proceed with calculations, we used a simple Lennard-Jones interaction potential for the fluid-solid interaction to obtain analytical estimates, although the main qualitative results do not depend on this hypothesis. We leave the detailed derivation in the Supplementary Information. A key prediction emerging from this calculation is that the friction coefficient is found to obey a simple functional dependence on the liquid molecular dynamics—characterized by its $\alpha$-peak frequency $f_\alpha$—in the general form

$$\lambda(f_\alpha) = \frac{\lambda_0}{2} \left( \frac{f_c}{f_\alpha} + \frac{f_\alpha}{f_c} \right) \tag{9}$$

with the two terms originating in the 'static' and 'dynamic' contributions to the interfacial friction, respectively. The expressions for $\lambda_0$ and $f_c$ as a function of the molecular parameters are provided in Supplementary Information.

As shown in Fig. 4, the prediction in Eq. (9) is in good agreement with the experimental data for the friction coefficient and its functional dependence on the molecular dynamics via $f_\alpha(T)$. The fit yields the values $f_c \approx 33.9$ MHz as the threshold frequency and $\lambda_0 \simeq 25.6 \cdot 10^6$ N.m/s. The scaling for the first (static) contribution, $\lambda^{stat} \propto 1/f_\alpha$, is expected for the friction on solids with a static roughness since the friction coefficient in this regime is merely proportional to the fluid relaxation time-scale, hence as $1/f_\alpha(T)$, in agreement with previous calculations[1]. This is consistent with the measured slip length being independent of the temperature for $T < 10$ °C, see Fig. 3a. In this regime, as sketched in Fig. 4b, the structure factors overlap is dominated by the solid's static corrugation at zero frequency, the influence of mica's phonons is thus negligible. At fast relaxation rates however, the scaling of the dynamical contribution to friction $\lambda^{dyn}(T) \propto f_\alpha(T)$ contradicts the Arrhenius picture and stems from the high overlap of liquid's structure factor $S_l$ with solid excitations $S_s^{dyn}$. The sketch of Fig. 4b highlights this crossover: above

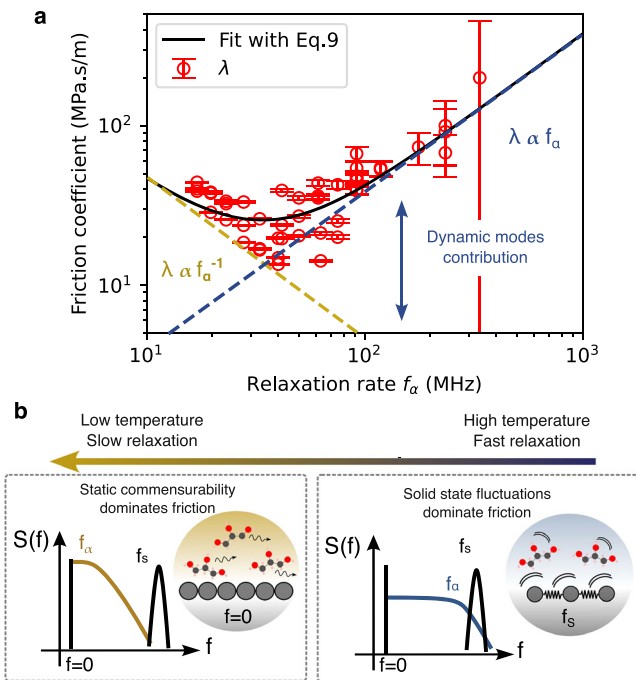

**Fig. 4 | Evidence of dynamical friction. a** We plot the experimental values of the solid−liquid friction coefficient, λ(T), versus the (temperature-dependent) frequency of the α-peak, $f_\alpha(T)$. The solid line is the theoretical prediction in Eq. (9). The fit to the experimental data yields $\lambda_0 = 25.6 \times 10^6$ N s/m$^3$ and $f_c = 33.9$ MHz. **b** Sketch of the underlying physical mechanisms, highlighting an increased dynamical dissipation when the fluid spectrum overlaps the solid internal spectrum. The liquid spectrum is described as a temperature-dependent Debye spectrum, with frequency $f_\alpha(T)$; cf Eq. (6). For the solid (black lines), the Dirac peak at zero frequency accounts for the static roughness of the solid, while the peak at $f_S$, characteristic of the solid excitations, is associated with the dynamical solid fluctuations. It is here modeled as a phonon spectrum; cf Eq. (8). The liquid-solid friction given by Eq. (5) depends on the overlap between the liquid and solid structure factors. Source data are provided as a Source Data file.

$f_c \sim 34$ MHz, the liquid's coupling with mica's phonons dominates that with the static corrugation at 0 Hz.

Altogether, the low and high-temperature regimes highlighted in the experimental data can therefore be interpreted as, respectively, the static (corrugation-induced) and dynamic (due to internal solid fluctuations) contributions to hydrodynamic friction. The cross-over between the two regimes occurs at a characteristic frequency $f_c$, which separates the static and dynamic frictional regimes.

The description therefore captures the main features of the experiments. However, we mention that estimating more quantitatively the measured values for $\lambda_0$ and $f_c$ (see above) on the basis on their predicted molecular expressions−as reported in the Supplementary Informations−is difficult because the latter depend very sensitively on various molecular details. It would also require to go beyond the simple theoretical description of the liquid and solid spectra, as well as the simplistic (Lennard-Jones type) molecular interactions that we use here. More fundamentally, the dynamics of supercooled liquids close to surfaces was shown to be strongly affected by surfaces, with drastic change of relaxation time-scales depending on the wall roughness[44,45]. This interfacial effect may affect the quantitative estimate of $f_c$ by orders of magnitudes, but it remains difficult to predict quantitatively. Hence we leave a full quantitative analysis for future work using proper molecular simulations and keep here our discussion to a qualitative−scaling−analysis which reproduces the main features of the experimental results.

As a last comment, the interfacial friction coefficient $\lambda_0$ on mica is measured to be very large (in the tens of MN.m/s), and only a viscous fluid−like glycerol here−could exhibit slippage on a surface with such high friction. This is consistent with the no slip boundary condition previously measured for water and OMCTS on mica[7,28].

In this work, we introduced a tuning-fork-based AFM optimized for liquid−solid friction measurement in a wide range of viscosity and temperature, both in vacuum and in a controlled atmosphere. We use this new system to explore the temperature-dependent slippage of supercooled glycerol on mica, hence providing an unprecedented insight into interfacial friction versus liquid's bulk dynamics. We not only evidence a massive increase of the slip length with decreasing temperature, but more unexpectedly a non-monotonous dependency of the friction coefficient on the liquid's relaxation rate.

In the low molecular frequency ($f_\alpha \rightarrow 0$) limit, we show that liquid friction decreases as $1/f_\alpha$, in good agreement with theoretical predictions for a frozen wall. On the other hand, above the threshold frequency $f_c \sim 34$ MHz, $\lambda$ is found to scale linearly with $f_\alpha$. We propose that a fluctuation-induced dissipation associated with the solid internal modes adds up to the corrugated-potential contribution. As liquid modes are blue shifted upon heating, they couple more efficiently with mica's phonons, which eventually overthrow the static corrugation contribution. Our theoretical analysis reproduces nicely the crossover between these static and dynamic frictional regimes, although a completely quantitative prediction is out of reach at this point. This picture echoes several recent works on friction and heat transfer[8,10,13,15,16], demonstrating that electronic and phononic excitations play a key role at liquid interfaces. Interestingly, internal wall dynamics are often omitted in molecular dynamics studies, our results show that they should in fact be given special attention[29,30].

On experimental grounds, the dynamical friction we demonstrate here offers a new tuning knob for liquid-friction control. In recent years, the careful engineering of phonon and electron band structures has become instrumental in the design of new thermoelectric materials. This could readily be applied to the specific engineering of liquid−solid interface properties, e.g. friction, thermal conduction or even mechano-electric energy conversion.

## Methods
### Materials
Glycerol was purchased from Sigma Aldrich. We verified by temperature-dependent viscosity measurements in a cone-plane rheometer that its purity was above 99.5% (see Supplementary Information). Muscovite mica disks were cleaved with scotch tape a few seconds only before the glycerol droplet was deposited on the surface and the vacuum chamber was pumped down to the mbar range.

### Experimental setup
The quartz tuning fork was purchased from RadioSpare and removed from its vacuum casing. A tungsten wire was glued with silver paste on the lower prong of the fork before it was etched between two liquid meniscii (2 M KOH solution) under a voltage drop of approximately 15 V. To ensure a spherical end, the tungsten tip was subsequently re-etched by applying the voltage drop between the wire itself and a single KOH meniscus. The tuning fork was then placed in a vacuum chamber on a 3 axis piezo-scanner having a 9 $\mu$m range (micro Tritor from PiezoJena). The scanner itself was placed on 3 axis inertial motors having centimetric range but nanometric steps (Mechonics). Finally, the mica sample was placed on a temperature-controlled sample holder, thermally connected to a Peltier element able to pump heat towards a heavy copper heatsink. A long working distance microscope was used for the initial coarse positioning of the tip on the sample.

### Environmental conditions
Right after the glycerol droplet was deposited on freshly cleaved mica, the vacuum chamber was pumped down to 1 mbar for several hours. In order to reduce the mechanical noise level, the chamber was then filled

with nitrogen, and the pumps were turned off before the tip was approached to the sample.

## Drainage flow measurements

The quartz tuning fork was excited mechanically by a piezo dither (Thorlabs) glued on the metallic holder. The fork's signal was directed to a phase-locked loop (PLL) electronic system allowing to lock the system at resonance (zero phase) and constant oscillation amplitude (e.g. 3 nm). From the frequency shift and excitation signals, we inferred the complex mechanical impedance of the glycerol drainage flow.

## Data availability

Full approach curves are available upon request to the corresponding authors. Source data are provided with this paper.

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

## Acknowledgements

L.B. acknowledges support by ERC project n-Aqua. B.C. acknowledges funding from a J.-P. Aguilar grant of the CFM Foundation. M.L. acknowledges funding by an AMX grant of Ecole Polytechnique. The authors thank N. Kavokine for many fruitful discussions.

## Author contributions

M.L., A.S., and L.B. coordinated the project. M.L. and G.M. performed the experiments and B.C. and M.L. developed the theoretical calculations with inputs from L.B. M.L, B.C., and L.B. discussed the results. M.L., B.C., and L.B. wrote the manuscript.

## Competing interests

The authors declare no competing interests.
