## [Peer Review File · Nature Communications]

REVIEWER COMMENTS

Reviewer #1 (Remarks to the Author):

Dear Editor,

The article Anomalous friction of supercooled glycerol on mica reports on unconventional, non-arrhenian friction behavior at the liquid-solid interface. As a model liquid the authors chose supercooled glycerol droplet deposited on mica surface. In their analysis the authors employ Atomic Force Microscope in tuning Fork configuration – the method is appropriate and commonly used in such type of studies. In the present study the AFM allowed to determine the slip length of glycerol on mica surface. The authors has found non-monotonous dependency of friction coefficient versus glycerol molecular dynamics and the relaxation rate of the glycerol molecules was controlled via temperature. The main message of the work is the important role of competition between static (corrugation induced) and dynamic (acoustic phonons induced) contributions to friction at the solid liquid interface. The results are supported with detailed calculations, which are very well presented mainly in the Supplementary Material.

The experiment is performed with care (the researchers precisely controlled the temperature as well as glycerol purity) and the findings are of general interest in the field of microfluidics and tribology. The article is written in very good language.

My minor comments to the authors is to improve on Figure 1. The scale bar on the SEM tip image is not visible. I would also suggests to increase slightly Figure 1c and explain with one sentence the slip length b in the figure caption.

In Figure 3 there is a typo: should be versus temperature T

My question to the authors:

1. Since the slip length b very much depends on the precision of determination of the zero-distance $D=0$, did the authors observed the saturation 'jump' of frequency shift signal in a reproducible manner for every spectra? Were the z-drift and the piezo creep negligible in that case?

2. I assume that glycerol axi-symmetric flow between AFM tip and mica surface depends on the tip oscillation amplitude (here $a=3\text{nm}$). How does the slip length depends on the oscillation amplitude. Could the authors comment on that? Few sentences about how the amplitude choice affects the flow would help. Possibly the authors have some results taken at different oscillation amplitudes and could include them into supplementary information.

Overall, the manuscript reads very well and the findings are novel, supported with serious calculations. I recommend the work for publication in Nature Communications after addressing my comments.

Reviewer #2 (Remarks to the Author):

Abdelhamid Maali Report on NCOMMS-24-19117:

I like this manuscript and I really enjoy reading it. I think that it is worthy of publication in Nature communication.

The key worthy element is that the authors are able to measure the temperature slip length of supercooled glycerol on mica, and have derived a simple theory for explaining their measurements.

I have a few comments that I would like to have clarification.

The cavitation effect.

Oscillating in liquid may create bubble that can deposit on the substrate, and which may affect the measurements. Can the authors explain whether such effect can occurs in their experiment and if any care was taken to avoid this.

Elastohydrodynamic effect

The authors discussed the effect of the mica deformation but they neglect the deformation of the glued tip. The Young modulus of the glue is much smaller than the mica. However, the interaction area on the mica is given by the hydrodynamic radius, and on the glue, the area is given by the area of the glued parts. I believe that this latter area can be orders of magnitude larger than the hydrodynamic radius and thus the elasto-hydrodynamic effect due to the glue may be neglected. I suggest to the authors to discuss this effect in the SI.

Frequency shift:

The quality factor of the oscillator decreases as the damping increases by orders of magnitude. I am not sure that the PLL will measure the dynamic resonance frequency of the cantilever for such high damping (low quality factor). In my point of view, the frequency shift measured by the PLL is not only related to the elastic part of the hydrodynamic interaction (it may be due to the damping). Furthermore, for high damping, the point mass model is not accurate to describe the oscillation frequency of the tuning fork. Using the parameters given in the manuscript and in the SI, I have calculated the frequency shift due to elasto-hydrodynamic interaction with mica, and I found that it increases and it reaches a value of several tens of hertz. However, the authors show a decrease of the quality factor. I think that the frequency shift can be used to define the contact position but it cannot be used to infer anything about the hydrodynamic interaction.

Minor suggestions:

- 1) The authors should give the value of the free quality factor.
- 2) It would be nice if the authors can add on figure Fig.2 a line that passes through the origin and that is parallel to the data at large distance (see the work of Charlaix PRL 2005).
- 3) In page 10 the densities n_l and n_s are not defined.

Typo errors:

- 1) Equation 4: missing (dt)
- 2) In SI, section viscosity measurement: the authors sometimes use $G(\omega)$ and sometimes $G(j\omega)$?
- 3) SI, last paragraph of viscosity measurements section they calculate the ratio between the two components of the viscosity at different values of the frequency.
- 4) In SI, Section Elasto-hydrodynamic corrections: the equation for D_k is not correct (the power $2/3$ is not in the right place).

In this manuscript, the authors report an anomalous temperature dependence of slip length for glycerol on mica. The experimental findings are interesting. However, the theoretical analysis is not solid. As a result, this manuscript still needs further revision before its publication. Here are details:

1. There have been series of studies on the temperature dependence on slip length. Many of them regard the boundary slip and liquid-solid friction as a rate-process, i.e. $l_s \sim \exp(E/T)$, with E being the activated energy. Here are examples:

[1] Liquid slip in nanoscale channels as a rate process. *Physical Review Letters* 2007, 98 (22), 226001.

[2] Temperature-controlled slip of polymer melts on ideal substrates. *Physical Review Letters* 2018, 121 (17), 177802.

[3] Temperature-dependent wall slip of Newtonian lubricants. *Journal of Fluid Mechanics* 2022, 948, 48.

[4] Temperature-dependent slip length for water and electrolyte solution. *Journal of Colloid and Interface Science* 2023, 636, 512-517.

In these studies, the liquid-solid energy barrier is regarded as the main factor that affect the boundary slippage. However, the effect of energy barrier is not mentioned in this manuscript. The authors should discuss why the common-used rate process theory is not applicable to the boundary slippage of glycerol in this manuscript and why mica is so special compared to the materials mentioned in the above materials.

2. Parts of theoretical analysis is confusing and seems to be self-contradictory. Here are examples:

2.1 The authors claim that the for the cases $T < 10^\circ\text{C}$, $\lambda \sim 1/f_\alpha$, which is inconsistent with Figure 3a in the manuscript where l_s is independent of T (page 13). However, according to the authors' definition: $f_\alpha = \exp(-E_a/k_B(T - T_g))$, which is temperature-dependent. So why $\lambda \sim 1/f_\alpha = \exp(E_a/k_B(T - T_g))$ is consistent with the result where l_s is independent of T ?

2.2 Similar to question 2.1, for cases $T > 10^\circ\text{C}$, the authors claim that $\lambda \sim f_\alpha(T)$ contradicts the Arrhenius picture (page 13). However, $f_\alpha = \exp(-E_a/k_B(T - T_g))$. Thus $\lambda \sim \exp(-E_a/k_B(T - T_g))$.

$k_B(T - T_g)$) is exactly the Arrhenius picture, showing no contradiction with Arrhenius.

2.3 There is no equation for dynamic structure factor S_{dyn} , is S_{ph} equivalent to S_{dyn} ?

2.4 The liquid structural factor S_l in main text (equation 6) and in SI is different (equation 18-20).

Where does the S_l come from?

2.5 Why the structural factors of solid can be expressed as equation 7 and 8? More details need to be supplemented.

2.6 What is the meaning for f_S ?

3. There are too many variables whose value is not mentioned.

3.1 The authors claim that the temperature dependence of slip length originate from phonon effects in this paper. The phonon effect in this paper is represented as S_{ph} . The magnitude comparison of S_{ph} and S_{stat} seems to represent the phonon effect. However, the calculation of S_{ph} and S_{stat} strongly relies on the calculation of water thickness $\delta(\omega)$. Expression and calculation $\delta(\omega)$ are not mentioned.

3.2 In addition, there are also constants not mentioned, such as: atomic density ρ_s , amplitude of roughness u_s , and Brillouin-zone-average q-structure $\bar{\phi}$ et al. A table for all these constant values and relevant references is needed.

3.3 Some key expressions lack references or deviations, such as: why $u_s^{ph} = T/mc^2$ (near equation 30 in SI)? why $c = \sqrt{B/\rho}$ (equation 31 in SI)? Other examples include equation (32, 33, 34)

4. There are too many approximations in the theoretical analysis. There are 10 independent “ \approx ” in theoretical derivation process. Even if the error in each approximation process is 10%, the accumulated total error is considerable ($0.9^{10}=34.8\%$). Will these approximations overestimate or underestimate the effects of phonons? This question needs to be discussed.

5. For experimental results at high temperature in Figure 3b and Figure 4a, the error bar is large. While for results at low temperature, the error bar is small. Experimental results with such error bar can be fitted by curves with many forms. Why the error of measurement depends on

temperature?

6. Recently, H. Ishida et al claimed that the classical boundary slip measurement method by O. Vinogradova and Cottin-Bizonne may introduce extra error (*International Journal of Thermofluids* 2024, 22, 100634). Considering the Vinogradova formula is key for the slip length estimation in the experiments, the accuracy of slip length measurement by AFM and SFA becomes a concern. The authors should discuss the accuracy of their measurement and analyze the possible errors in details.
7. Based on the above problems 2, 3, 4, 5 and 6, attributing the temperature dependence of slip length to phonon effect is questionable. Too many effects can bring variations and errors for slip length measurement.
8. This experiment is conducted in glycerol for convenience. Are the results only special cases for mica and glycerol? Are the results applicable for other liquid-solid interfaces? The universality of experimental and theoretical results should be discussed.

We thank the three reviewers for their insightful comments. We have made a thorough revision of the manuscript following their remarks. We answer below their questions, point by point.

The answer to the reviewers' question are in orange. And the modifications to the manuscript and Supplementary Informations are highlighted in blue, as well as in the corresponding updated versions.

Reviewer #1

The article Anomalous friction of supercooled glycerol on mica reports on unconventional, non-arrhenian friction behavior at the liquid-solid interface. As a model liquid the authors chose supercooled glycerol droplet deposited on mica surface. In their analysis the authors employ Atomic Force Microscope in tuning Fork configuration – the method is appropriate and commonly used in such type of studies. In the present study the AFM allowed to determine the slip length of glycerol on mica surface. The authors has found non-monotonous dependency of friction coefficient versus glycerol molecular dynamics and the relaxation rate of the glycerol molecules was controlled via temperature. The main message of the work is the important role of competition between static (corrugation induced) and dynamic (acoustic phonons induced) contributions to friction at the solid liquid interface. The results are supported with detailed calculations, which are very well presented mainly in the Supplementary Material.

The experiment is performed with care (the researchers precisely controlled the temperature as well as glycerol purity) and the findings are of general interest in the field of microfluidics and tribology. The article is written in very good language.

My minor comments to the authors is to improve on Figure 1. The scale bar on the SEM tip image is not visible. I would also suggests to increase slightly Figure 1c and explain with one sentence the slip length b in the figure caption. In Figure 3 there is a typo: should be versus temperature

We thank the reviewer for their positive comments on our experiments. We have followed their advice regarding Figure 1. We have increased the size of the SEM image and move it to the SI (Figure S5a). We also added a comment on the sketch of Fig 1c.

My question to the authors:

1. Since the slip length b very much depends on the precision of determination of the zero-distance $D=0$, did the authors observed the saturation 'jump' of frequency shift signal in a reproducible manner for every spectra? Were the z-drift and the piezo creep negligible in that case?

Indeed, the reviewer is right in pointing out that the determination of solid-contact position is important for the measurement. We have indeed checked carefully that the discontinuity in the frequency-shift signal is measured in a systematic and perfectly reproducible manner. To illustrate this point, we show below several other examples showing how the discontinuity occurs systematically (here for various temperatures).

Figure 1 : **Solid contact detection for an oscillation amplitude of 3 nm and temperatures ranging from -0.6 to 41°C** The black line denotes a threshold frequency shift which is chosen independently for each curve in order to optimize the precision on the contact's determination.

We have added these curves in the supplementary information (Figure S3) along with a sentence in the main manuscript to emphasize this point.

2. I assume that glycerol axi-symmetric flow between AFM tip and mica surface depends on the tip oscillation amplitude (here $a=3\text{nm}$). How does the slip length depends on the oscillation amplitude. Could the authors comment on that? Few sentences about how the amplitude choice affects the flow would help. Possibly the authors have some results taken at different oscillation amplitudes and could include them into supplementary information.

We have indeed checked the (non-) influence of the oscillation amplitude. One expects in the linear regime that the mechanical impedance Z is independent of amplitude, as well as the slip length. We report below measurements of approach curves $Z(D)$ at 25°C for various oscillation amplitudes, ranging from $a=1$ to $a=5$ nm. These measurements confirm that the slip length b is insensitive to the oscillation amplitude within error bars (here $b = 3.5 \pm 1$ nm).

Following the reviewer suggestion, we have added a comment about this point in the main text and reported the curves (below) in the supplementary informations (SI-figure S4).

Figure 2: **Dependency of approach curves on the tuning fork's oscillation amplitude at 25°C** We plot the frequency shift δf with the corresponding solid contact's position with blue curves and the inverse dissipative impedance $1/\text{Im}(Z)$ with golden points. The black lines are fits with the Reynolds equation. Panels a-c correspond respectively to oscillation amplitudes of 1, 3 and 5 nm.

Overall, the manuscript reads very well and the findings are novel, supported with serious calculations. I recommend the work for publication in Nature Communications after addressing my comments.

We thank again the referee for their positive feedbacks on our work.

Reviewer #2

Abdelhamid Maali Report on NCOMMS-24-19117:

I like this manuscript and I really enjoy reading it. I think that it is worthy of publication in Nature communication. The key worthy element is that the authors are able to measure the temperature slip length of supercooled glycerol on mica, and have derived a simple theory for explaining their measurements. I have a few comments that I would like to have clarification.

We thank the reviewer for his very positive feedback on the manuscript and his suggestions to improve it, which we carefully took into account in the revised manuscript.

The cavitation effect.

Oscillating in liquid may create bubble that can deposit on the substrate, and which may affect the measurements. Can the authors explain whether such effect can occurs in their experiment and if any care was taken to avoid this.

We thank the reviewer for this interesting question. As we discuss below, in view of our experimental results, we expect that cavitation does not occur in our experiments. Indeed, a cavitation event, if any, would induce dramatic and abrupt changes in the mechanical response on both the elastic and dissipative parts. In particular the onset of cavitation would generate a strong additional dissipative response. We do not observe such a behavior and considering the smooth and continuous nature of our curves, we can argue that the threshold to cavitation is not reached in our experiments.

Note also that under our conditions, the pressure variations in the liquid meniscus are at most in the bar range, even for a strongly viscous liquid like glycerol (as can be estimated from the theoretical expectations for the pressure profile in the present geometry). Such values are much smaller than usually measured cavitation thresholds. For example, existing measurements in the literature for water point show that the cavitation threshold for water is at least in the minus tens of MPa (see e.g. F. Caupin, E. Herbert, C. R. Physique 7 (2006) 1000–1017 and all recent literature on this topics, also by Caupin et al.). Furthermore, the excellent wetting of glycerol on both the tungsten tip and the mica surface ensures that gas bubbles will not be preferentially nucleated at surfaces with respect to the bulk liquid (homogenous case).

For these reasons, we are confident that cavitation is not playing a role in our experiments.

Elastohydrodynamic effect. The authors discussed the effect of the mica deformation but they neglect the deformation of the glued tip. The Young modulus of the glue is much smaller than the mica. However, the interaction area on the mica is given by the hydrodynamic radius,

and on the glue, the area it is given by the area of the glued parts. I believe that this latter area can be orders of magnitude larger than the hydrodynamic radius and thus the elasto-hydrodynamic effect due to the glue may be neglected. I suggest to the authors to discuss this effect in the SI.

We thank the reviewer for this comment. Indeed, elasto-hydrodynamic effects are drastically augmented by the softness of the glue but considering the surface of the glued part is larger than the hydrodynamic area by a factor 10^5 : $S_{glue} \sim 10^{-8}m^2 \sim 10^5Rh$. In this context, the elasto-hydrodynamic response of the glue is completely negligible as compared to that of the tip and substrate.

Following the reviewer suggestion, we added a discussion of this effect in the SI (II.E).

Frequency shift: The quality factor of the oscillator decreases as the damping increases by orders of magnitude. I am not sure that the PLL will measure the dynamic resonance frequency of the cantilever for such high damping (low quality factor). In my point of view, the frequency shift measured by the PLL is not only related to the elastic part of the hydrodynamic interaction (it may be due to the damping). Furthermore, For high damping, the point mass model is not accurate to describe the oscillation frequency of the tuning fork. Using the parameters given in the manuscript and in the SI, I have calculated the frequency shift due to elasto-hydrodynamic interaction with mica, and I found that it increases and it reaches a value of several tens of hertz. However, the authors show a decrease of the quality factor. **I think that the frequency shift can be used to define the contact position but it cannot be used to infer anything about the hydrodynamic interaction.**

We thank the reviewer for this remark and we actually fully agree with this comment. We further stress that this comment has no impact on our study as the frequency shift's divergence is used only as a means to detect the contact position.

We have suppressed the corresponding discussion in our manuscript in order to avoid any confusion or misunderstanding.

Minor suggestions:

1) The authors should give the value of the free quality factor.

The free quality factor of our resonator is a few thousands but drops to roughly 100 at long distance when the tip is immersed in glycerol. Such values of Q allow an excellent operation of the phase locked-loop for phase and amplitude regulations and thus precise measurements of the complex mechanical impedance Z.

We have added the long-range value of Q on page 4 of the manuscript.

2) It would be nice if the authors can add on figure Fig.2 a line that passes through the origin and that is parallel to the data at large distance (see the work of Charlaix PRL 2005).

We thank the referee for this remark. We considered carefully this suggestion but finally judged that it would make the Figure too complicated and be detrimental to the clarity of the manuscript. Thus, we left Figure 2 in its initial state.

3) In page 10 the density n_l and n_s are not defined.

We thank the reviewer for pointing out this omission. These quantities account for the liquid and solid density distributions. They are now defined (on page 11 of the main text).

Typo errors:

1) Equation 4: missing (dt)

We have corrected this typo and thank the reviewer for pointing it out.

2) In SI, section viscosity measurement: the authors sometimes use $G(w)$ and sometime $G(jw)$?

Indeed, they denote the same observable but we agree with the referee that this notation is confusing.

Accordingly, we modified the notations back to $G(\omega)$ to avoid any confusion.

3) SI, last paragraph of viscosity measurements section they calculate the ratio between the two components of the viscosity at different value of the frequency.

This was indeed a typo, the ratio is that of the real and imaginary parts of viscosity, both taken at 30 kHz. We have now corrected it.

4) In SI, Section Elastohydrodynamic corrections: the equation of D_k is not correct (the power $2/3$ is not at right place.

We thank the reviewer for signaling this typo and have corrected it in the new version.

Reviewer #3

In this manuscript, the authors report an anomalous temperature dependence of slip length for glycerol on mica. The experimental findings are interesting. However, the theoretical analysis is not solid. As a result, this manuscript still needs further revision before its publication. Here are details:

1. There have been series of studies on the temperature dependence on slip length. Many of them regard the boundary slip and liquid-solid friction as a rate-process, i.e. $l \sim \exp(E/T)$, with E being the activated energy. Here are examples:

*[1] Liquid slip in nanoscale channels as a rate process. **Physical Review Letters** 2007, 98 (22), 226001.*

*[2] Temperature-controlled slip of polymer melts on ideal substrates. **Physical Review Letters** 2018, 121 (17), 177802.*

*[3] Temperature-dependent wall slip of Newtonian lubricants. **Journal of Fluid Mechanics** 2022, 948, 48.*

[4] Temperature-dependent slip length for water and electrolyte solution. **Journal of Colloid and Interface Science** 2023, 636, 512-517.

In these studies, the liquid-solid energy barrier is regarded as the main factor that affect the boundary slippage. However, the effect of energy barrier is not mentioned in this manuscript. The authors should discuss why the common-used rate process theory is not applicable to the boundary slippage of glycerol in this manuscript and why mica is so special compared to the materials mentioned in the above materials.

We thank the reviewer for the comments and the references provided. We have modified the manuscript and incorporated those of the suggested references which were not yet cited.

However, we do not fully understand and actually respectfully disagree about the assessment that the *'theoretical analysis is not solid'*. These are strong words which do not reflect the theoretical analysis performed, and its success to describe the experimental data. The previous text may have been not clear enough, but the analysis followed all the rigor required to describe the friction process, a topic on which we have a long-standing experience.

It also does not reflect the considerable efforts on the experimental side to perform these challenging experiments. Theory advances thanks to experiments. Here the experiments point to an unexpected behavior, and the proposed theory, which is new to our knowledge, predicts scaling behaviors which are able to account for the experimental behaviors.

We emphasize accordingly that the proposed analysis is motivated by the experimental results themselves, which cannot be explained by 'standard' descriptions. In particular – and this may have been not sufficiently emphasized in the previous version of the manuscript -, our experiments show that the friction coefficient *cannot be accounted for by an Arrhenius/activation behavior*.

Indeed, our experimental results show an increase of the friction coefficient with temperature at 'large' temperatures. This is in contrast with an Arrhenius description, $\lambda \sim \exp[E_a/k_B T]$, which would predict – as expected - a *decrease of friction with temperature*. The latter behavior would be predicted by a simple barrier crossing argument and it has indeed been verified in molecular dynamics simulations without wall excitations, see eg Lafon et al., figure 3 of our Ref [30]. But this behavior is in contrast with the experimental observations. Our analysis then points to a mechanism based on the internal excitations of the solid, which accounts qualitatively and semi-quantitatively for the experimental results. To our knowledge, this mechanism was not previously discussed (even using molecular dynamics simulations).

However, following the suggestion by the referee and in order to clarify our discussions, we have added a paragraph to discuss our results in the framework of the rate process theory on page 9.

2.

Parts of theoretical analysis is confusing and seems to be self-contradictory. Here are examples:

2.1 The authors claim that for the cases $T < 10^\circ\text{C}$, $\lambda \sim 1/f_\alpha$, which is inconsistent with Figure 3a in the manuscript where l_s is independent of T (page 13). However, according to the authors' definition: $f_\alpha = \exp(-E_\alpha/k_B(T - T_g))$, which is temperature-dependent. So why $\lambda \sim 1/f_\alpha = \exp(E_\alpha/k_B(T - T_g))$ is consistent with the result where l_s is independent of T ?

The low temperature behavior for the friction coefficient, *i.e.* $\lambda \sim 1/f_\alpha$ is actually fully consistent with the *slip length* being mostly independent of T at these 'small' temperatures, as we now clarify.

First, we emphasize that the temperature dependency of the friction coefficient $\lambda(T) = \eta(T)/b(T)$ is radically different from that of the slip length b itself (which is denoted as l_s by the reviewer), because of the strong dependency of viscosity η on temperature (see SI).

This temperature independence of the slip length in the low temperature regime can then be understood based on a simple estimate of $b(T)$. In this regime, one expects the internal excitations of the solid to be negligible. According to the Green-Kubo expression for the friction coefficient (Eq. 4), one has typically $\lambda \sim \langle F^2 \rangle \times \tau_\alpha \sim 1/f_\alpha$, with $\tau_\alpha = 1/f_\alpha$ the typical relaxation time of the liquid. We note that such behavior has been amply verified by molecular dynamics simulations with frozen walls (see eg Falk et al. NanoLetters 10 4067 (2010), now ref. [42], also Herrero et al. our ref [29]). Now an estimate for the viscosity of the fluid itself is typically given as $\eta = G \times \tau_\alpha \sim 1/f_\alpha$, with G the high-frequency elastic modulus. This shows that the viscosity and friction coefficient vary both as $1/f_\alpha$. Accordingly, their ratio, which is precisely the slip length, becomes independent of the characteristic time or frequency of the liquid, hence $b(T)$ varies little with the temperature in this regime. This is a scaling estimate, which gives a rough estimate, but this is indeed what is observed in our experiments. Altogether, this shows that in this 'low temperature' regime, the slip length does not vary much with the temperature, in strong contrast to liquid characteristic frequency f_α .

2.2 Similar to question 2.1, for cases $T > 10^\circ\text{C}$, the authors claim that $\lambda \sim f_\alpha(T)$ contradicts the Arrhenius picture (page 13). However, $f_\alpha = \exp(-E_\alpha/k_B(T - T_g))$. Thus $\lambda \sim \exp(-E_\alpha/k_B(T - T_g))$ is exactly the Arrhenius picture, showing no contradiction with Arrhenius.

Following our arguments above, we respectfully disagree with the statement that $\lambda \sim f_\alpha$, as observed in the high temperature regime, is in agreement with the Arrhenius picture.

First we agree with the referee that the characteristic frequency/inverse time $f_\alpha = 1/\tau_\alpha$ will indeed behave like $f_\alpha \sim \exp[-E_\alpha/k_B T]$ (with a possible T_g correction in the temperature) and this frequency increases with the temperature.

But this differs from the Arrhenius picture for the friction coefficient, which predicts that the friction coefficient will behave as $\lambda \sim \exp[+E_\alpha/k_B T]$. Actually in such activation picture, one would expect the friction coefficient to be proportional to the crossing-time of the barrier, *i.e.*

in agreement with the previous estimate discussed above: $\lambda \sim \langle F^2 \rangle \times \tau_\alpha \sim 1/f_\alpha$. This is the 'naïve' expectation for the friction coefficient, with a small barrier crossing times leading to small dissipation and this activation prediction does not account for the solid's internal excitations.

Altogether, this "Arrhenius" argument would suggest that the friction coefficient λ would scales as $\lambda \sim \exp[+E_\alpha/k_B T]$, so that the friction coefficient would decrease with temperature. And, as we discuss above and in the manuscript, this "Arrhenius" behavior is typically what is observed in molecular dynamics simulations assuming 'frozen' confining walls which does not exhibit collective excitations, see eg Lafon et al., our Ref [30].

But our experimental observations at 'large' temperature are in strong disagreement with this prediction. Hence this points that the Arrhenius picture does not account for the experimental results and does originate in a novel phenomenon for which a new mechanism has to be proposed. This is what our theory proposes.

We agree that the Arrhenius terminology may lead to some confusion in the wording and we have rephrased the corresponding sentences in the manuscript. A paragraph has been rewritten on page 9.

2.3 There is no equation for dynamic structure factor S_{dyn} , is S_{ph} equivalent to S_{dyn} ?

We thank the reviewer for pointing out this omission. Indeed, the solid's dynamical structure factor is a composite quantity made of the various thermally populated excitations of the solid surface (including in general *e.g.* phononic and electronic modes – for conducting systems -). In the description, we approximate S_{dyn} by a single acoustic phonon branch S_{ph} which allows to calculate a theoretical estimate of the dynamical contribution to friction.

Following the reviewer's comment, we have clarified this point in the manuscript, in particular in Eq. (8).

2.4 The liquid structural factor S_l in main text (equation 6) and in SI is different (equation 18-20).

Where does the S_l come from?

S_l is the liquid's dynamical structure factor, describing the density fluctuations in the liquid. Considering the characteristic Debye shape in dielectric spectra reported by Lunkenheimer *et al.* [3], we assume such a shape for S_l in Eq.6 of the main text. We stress that this expression (Eq.6) is valid a priori in the long wavelength limit ($q \rightarrow 0$). At higher wavevectors, no analytical expression has been reported for the dynamical structure factor of glycerol, but like for all liquids, it is expected to decay at a few molecular scales with a functional dependency $\phi(q)$. The latter is not known precisely, but its precise shape is not essential to obtain the scaling behaviors for the friction coefficient.

Following the reviewer's comment, we corrected Eq.SI 18 where $f(q)$ was used instead of $\phi(q)$ by mistake.

2.5 Why the structural factors of solid can be expressed as equation 7 and 8? More details need to be supplemented.

The structural factors of the solid are respectively S_{stat} and S_{dyn} for the static (zero frequency) and dynamical (non-zero frequency) components.

Static structure factor: The static structure factor of the crystalline surface is computed as the Fourier transform of the correlations of atomic density $\rho(\mathbf{r}, t)$:

$$S_{stat}(\mathbf{q}) = \int d^2r \int_0^{+\infty} dz e^{-i\mathbf{q}\mathbf{r}-qz} \langle \rho(R, t) \rho(\mathbf{r} + R, t) \rangle_{R,t}$$

Equivalently, it may be computed from the surface radial distribution function $g(r)$:

$$S_{stat}(q) = \int_0^{+\infty} dr 4\pi r^2 \text{sinc}(qr) (g(r) - g(\infty))$$

General structure factor: Similarly, the total structure factor is defined as:

$$S_s(\mathbf{q}, \omega) = \int dt d^2r \int_0^{+\infty} dz e^{-i\mathbf{q}\mathbf{r}-qz+i\omega t} \langle \rho(R, T) \rho(\mathbf{r} + R, T + t) \rangle_{R,T}$$

The general structure factor can accordingly be written as the decomposition:

$$S_s(\mathbf{q}, \omega) = S_{stat}(\mathbf{q}) \delta(\omega) + S_{dyn}(\mathbf{q}, \omega)$$

separating the static “ $\omega = 0$ ” contribution from the non-zero frequency contribution.

The Dirac distribution $\delta(\omega)$ (also in Eq.7 of the manuscript) allows to isolate the zero-frequency component fixed by the crystalline structure of mica, which corresponds to the static structure factor. The latter static structure factor will exhibit some molecular feature of the liquid structure. Typically, it will exhibit a peak around a \mathbf{q} vector associated with the molecular length scale $q_a^{max} = 2\pi/\sigma_s$ (with a molecular length scale σ_s). We note that this structure and this peak is responsible for the corresponding ‘roughness-induced’ friction, as shown *e.g.*, in the paper by Falk et al. now cited in the revised manuscript as Ref [42].

Conversely, the dynamical contribution to the structure factor, which we denote here as $S_{dyn}(\mathbf{q}, \omega)$, corresponds to all modes of the solid at non-zero frequency. Here, we only consider a phonon mode and thus make, as discussed in question 2.3, the assumption that $S_{dyn} \sim S_{ph}$. Finally, in Eq.8, the Dirac function $\delta(\omega - qc)$ yields the resonant condition of the acoustic surface phonon which is a simple model to describe the solid phonon’s dynamics.

We have now added a comment to explain the static structure factor, its link to the roughness-induced friction – with the added reference [42] - and clarified the expression for the dynamic structure factor in Eq. (8).

2.6 What is the meaning for f_S ?

The frequency f_S on Figure 4 denotes the frequency of a model solid mode (black peak). This single-mode picture is used to clarify the resonance condition between one typical solid mode and the liquid's Debye peak. In the subsequent theoretical computations, we used a more realistic acoustic phonon branch (with a linear dispersion) to describe the dynamics of mica's surface. The notation f_S is used as a generic notation for the "solid" peak.

We accordingly clarified the caption of Figure 4, and the meaning of f_S .

3.

There are too many variables whose value is not mentioned.

3.1 The authors claim that the temperature dependence of slip length originate from phonon effects in this paper. The phonon effect in this paper is represented as S_{ph} . The magnitude comparison of S_{ph} and S_{stat} seems to represent the phonon effect. However, the calculation of S_{ph} and S_{stat} strongly relies on the calculation of water thickness $\delta(\omega)$. Expression and calculation $\delta(\omega)$ are not mentioned.

We thank the reviewer for pointing this issue, which was indeed confusing in the previous version of the manuscript. There is actually a confusion on the meaning of δ : between the Dirac distribution accounting for the static modes at zero frequency, and the water layer thickness, say δ_{water} .

We have made the required modifications in the revised version. In order to clarify the notations, we have now corrected the manuscript on page 10 and replaced δ by δ_{water} when it denotes a possible water layer's thickness. We have also clarified the meaning of $\delta(\omega)$ as Dirac distribution after Eq (7).

3.2 In addition, there are also constants not mentioned, such as: atomic density ρ_s , amplitude of roughness u_s , and Brillouin-zone-average q-structure $\bar{\phi}$ et al. A table for all these constant values and relevant references is needed.

We thank the reviewer for this comment and we added a table of the relevant parameters in the Supplementary Information.

Table I on page 8 of the Supplementary Information gathers the most important parameters of the model and their numerical estimates in the relevant case.

3.3 Some key expressions lack references or deviations, such as: why $u_s^{ph} = T/mc^2$ (near equation 30 in SI)? why $c = \sqrt{B/\rho}$ (equation 31 in SI)? Other examples include equation (32, 33, 34)

3.3 We thank the reviewer for pointing this out. The typical phonon amplitude, in units of the atomic distance σ_s is estimated by comparing thermal energy to the phonon mode's kinetic energy. The expression for the phonon's velocity can be found in standard crystal physics textbook (see for example the book by Ashcroft-Mermin).

Following the reviewer's comments, we now added more details on the derivation of the phonon model after Eq.30 of the SI. A prefactor k_B was also missing in front of the temperature.

4. There are too many approximations in the theoretical analysis. There are 10 independent " \approx " in theoretical derivation process. Even if the error in each approximation process is 10%, the accumulated total error is considerable ($0.9^{10}=34.8\%$). Will these approximations overestimate or underestimate the effects of phonons? This question needs to be discussed.

We thank the reviewer for pointing out this appreciation.

We emphasize that our objective is to explain the anomalous behavior highlighted in our experiments, which cannot be explained by 'standard' arguments. Accordingly, we predict two regimes with different scaling behaviors for the friction coefficient in terms of the liquid characteristic time f_α . Our analysis aims at a *scaling prediction*, and the \sim sign is justified in this respect.

In particular, we aim at finding an order of magnitude of the dynamical contribution to friction but by no means a fully quantitative derivation. The reason is simply that a full description would require a precise knowledge of the dynamical structure factors of both glycerol and mica, while we only know these quantities in a qualitative way. For instance, the q -dependency of dynamical structure factors $S_l(q, \omega)$ and $S_{dyn}(q, \omega)$ is very hard to obtain by spectroscopy techniques and this is why we resort to several approximations and do not claim a fully quantitative calculation.

On the other hand, a quantitative agreement would require dedicated molecular dynamics simulations including the solid degrees of freedom. This goes far beyond the present – mostly experimental – paper. But it will certainly motivate the community to explore such leads !

5. For experimental results at high temperature in Figure 3b and Figure 4a, the error bar is large. While for results at low temperature, the error bar is small. Experimental results with such error bar can be fitted by curves with many forms. Why the error of measurement depends on temperature?

In our experimental protocol, we do measure the slip length as a function of temperature $b(T)$ and then compute the liquid-solid friction coefficient from the expression $\lambda = \eta/b$. The experimental error on λ is obtained by propagating the error on b according to: $\Delta\lambda = \eta\Delta b/b^2$. Now considering the vanishing of b at high temperature and a constant error on b , we expect a large increase of the error on λ upon heating as $1/b(T)^2$.

6. Recently, H. Ishida et al claimed that the classical boundary slip measurement method by O. Vinogradova and Cottin-Bizonne may introduce extra error (*International Journal of Thermofluids* 2024, 22, 100634). Considering the Vinogradova formula is key for the slip length estimation in the experiments, the accuracy of slip length measurement by AFM and SFA becomes a concern. The authors should discuss the accuracy of their measurement and analyze the possible errors in details.

We acknowledge that this paper suggests an interesting analysis to improve the accuracy of the slip length measurement. However, we note that the systems under consideration are not similar. While this paper investigates the errors using a standard AFM measurement, we use a different *Tuning-Fork* AFM instrument. Our system allows us to allay the limitations associated with the 'recursive method' in the classical AFM - quoting the denomination of the above paper -. In particular we are not limited by the measurement of the spring constant, nor the local heating induced by the laser, which may occur in standard AFM. Finally, there is no real issue with the description by Vinogradova, rather its use in the analysis using standard AFM. In our case, the description by Vinogradova is fully sufficient to obtain the slip lengths with an experimental error in the nanometer range.

7. Based on the above problems 2, 3, 4, 5 and 6, attributing the temperature dependence of slip length to phonon effect is questionable. Too many effects can bring variations and errors for slip length measurement.

We believe that one should come back to the essence of our study, which is mostly experimental, with a solid and new theoretical interpretation. As acknowledged by the other referees, the experiments were performed with all due care and checks in order to obtain a clean and robust measurement of the friction coefficient with temperature. The experiments demonstrate an unconventional behavior for the friction coefficient, in the form of a non-monotonous behavior with temperature. This is a new result and the key result of the paper. And this behavior cannot be explained in terms of the existing theoretical framework, nor existing molecular dynamics simulations assuming frozen walls (as we discuss in the paper).

This shows that another ingredient and mechanism has to be introduced in order to rationalize the experimental behavior. We show that introducing the internal degrees of freedom of the solid allows to account for the anomalous behavior of the friction coefficient. This takes the form of *a scaling behavior of the friction versus the liquid characteristic frequency f_α* , with two regimes exhibiting two different scaling behaviors for the friction. The theoretical predictions are in agreement with the experimental results for the scaling behavior of the friction. Again, this is the key result of our work. Phonon modes are naturally expected to exist in the solid, and the prediction of the phononic friction is in excellent qualitative agreement with the non-monotonic behavior we report. Our theoretical analysis shows a semi-quantitative agreement with experiments.

This is the main message of the paper. And we hope and foresee that this will generate a surge of interest, in particular for computational models, in order to explore our findings.

8. This experiment is conducted in glycerol for convenience. Are the results only special cases for mica and glycerol? Are the results applicable for other liquid-solid interfaces? The universality of experimental and theoretical results should be discussed.

We have chosen the mica-glycerol pair as a model and generic system for liquid-solid friction, and we expect the results to be fully generalizable as we now explain. Firstly, the atomic smoothness of mica has made it the model surface for experiments, being universally used for liquid confinement since the early days of the SFA.

Now turning to glycerol, it has been used as a model liquid for the study of glass transition since decades. Beyond mere convenience, glycerol is an excellent model liquid for liquid-friction measurements for various reasons:

- As a simple molecular liquid, it does not show any of the long-chain phenomenology of polymers.
- Its H-bonds network is coupling the translational and rotational degrees of freedom of the liquid, allowing to infer local translational motions from the dielectric spectroscopy, the response to which is dominated by rotations. [1-3]
- It is a H-bound liquid and thus has a large number of common features with water. For these reasons, it is also used as a model system in biology and is widely used in combination with water as an anti-freezing agent both in industry and by living organisms. [2]

Finally, the mica-glycerol interface is chemically inactive. Mica being neither very glycerol-philic nor glycerol-phobic, we argue that this interface is an excellent model system, in direct line of several decades of investigations of liquids on mica. Overall, our experiments truly have a vocation to unveil general effects of the liquid-solid interface.

[1] Sippel, P., S. Krohns, D. Reuter, P. Lunkenheimer, et A. Loidl. « Importance of Reorientational Dynamics for the Charge Transport in Ionic Liquids ». *Physical Review E* 98, n° 5 (15 novembre 2018): 052605.

[2] González, J. A. T., Longinotti, M. P., & Corti, H. R. (2012). Viscosity of supercooled aqueous glycerol solutions, validity of the Stokes–Einstein relationship, and implications for cryopreservation. *Cryobiology*, 65(2), 159-162.

[3] Lunkenheimer, P., Schneider, U., Brand, R., & Loidl, A. J. C. P. (2000). Glassy dynamics. *Contemporary Physics*, 41(1), 15-36.

REVIEWERS' COMMENTS

Reviewer #1 (Remarks to the Author):

Dear Editor,

The authors of the manuscript entitled Anomalous friction of supercooled glycerol on mica have addressed all my comments in detail. Overall, the changes to the main text and supplementary materials were made carefully and improved the paper. I therefore recommend publication of the manuscript in Nature Communications.

Yours faithfully

Reviewer #2 (Remarks to the Author):

The authors have corrected the manuscript according my suggestions and have removed my concerns. I recommend the manuscript for publication in Nature Communications.

Reviewer #3 (Remarks to the Author):

The authors have addressed all my concerns.